# SafeNET: Initial development and validation of a real-time tool for predicting mortality risk at the time of hospital transfer to a higher level of care

Stefanie C. Altieri Dunn[1], Johanna E. Bellon[1], Andrew Bilderback[1], Jeffrey D. Borrebach[1], Jacob C. Hodges[1], Mary Kay Wisniewski[1], Matthew E. Harinstein[2], Tamra E. Minnier[1], Joel B. Nelson[3], Daniel E. Hall[1,4,5]*

1 The Wolff Center at UPMC, Pittsburgh, Pennsylvania, United States of America, 2 Heart and Vascular Institute, University of Pittsburgh Medical Center, Pittsburgh, Pennsylvania, United States of America, 3 Department of Urology, University of Pittsburgh School of Medicine, Pittsburgh, Pennsylvania, United States of America, 4 Department of Surgery, University of Pittsburgh, Pittsburgh, Pennsylvania, United States of America, 5 Center for Health Equity Research and Promotion, VA Pittsburgh Healthcare System, Pittsburgh, Pennsylvania, United States of America

* hallde@upmc.edu

## Abstract

### Background

Processes for transferring patients to higher acuity facilities lack a standardized approach to prognostication, increasing the risk for low value care that imposes significant burdens on patients and their families with unclear benefits. We sought to develop a rapid and feasible tool for predicting mortality using variables readily available at the time of hospital transfer.

### Methods and findings

All work was carried out at a single, large, multi-hospital integrated healthcare system. We used a retrospective cohort for model development consisting of patients aged 18 years or older transferred into the healthcare system from another hospital, hospice, skilled nursing or other healthcare facility with an admission priority of direct emergency admit. The cohort was randomly divided into training and test sets to develop first a 54-variable, and then a 14-variable gradient boosting model to predict the primary outcome of all cause in-hospital mortality. Secondary outcomes included 30-day and 90-day mortality and transition to comfort measures only or hospice care. For model validation, we used a prospective cohort consisting of all patients transferred to a single, tertiary care hospital from one of the 3 referring hospitals, excluding patients transferred for myocardial infarction or maternal labor and delivery. Prospective validation was performed by using a web-based tool to calculate the risk of mortality at the time of transfer. Observed outcomes were compared to predicted outcomes to assess model performance.

The development cohort included 20,985 patients with 1,937 (9.2%) in-hospital mortalities, 2,884 (13.7%) 30-day mortalities, and 3,899 (18.6%) 90-day mortalities. The 14-

**Data Availability Statement:** The dataset for this human subjects research cannot be feasibly de-identified because it contains elements of dates

and other protected health information that could be reassembled to identify subjects. Furthermore, the data are subject to HIPAA authorization that cannot be obtained, and they are held under the covered entity of UPMC. Per UPMC policy, disclosure of data would require suitable Data Use Agreement and this can be pursued on request. The UPMC Quality Review Committee can be reached at AskQRC@upmc.edu.

**Funding:** The authors received no specific funding for this work.

**Competing interests:** The authors have declared that no competing interests exist.

variable gradient boosting model effectively predicted in-hospital, 30-day and 90-day mortality (c = 0.903 [95% CI:0.891–0.916]), c = 0.877 [95% CI:0.864–0.890]), and c = 0.869 [95% CI:0.857–0.881], respectively). The tool was proven feasible and valid for bedside implementation in a prospective cohort of 679 sequentially transferred patients for whom the bedside nurse calculated a SafeNET score at the time of transfer, taking only 4–5 minutes per patient with discrimination consistent with the development sample for in-hospital, 30-day and 90-day mortality (c = 0.836 [95%CI: 0.751–0.921], 0.815 [95% CI: 0.730–0.900], and 0.794 [95% CI: 0.725–0.864], respectively).

## Conclusions

The SafeNET algorithm is feasible and valid for real-time, bedside mortality risk prediction at the time of hospital transfer. Work is ongoing to build pathways triggered by this score that direct needed resources to the patients at greatest risk of poor outcomes.

## Introduction

Each year, nearly 1.6 million patients are transferred to referral centers accounting for as much as 3.5% of all inpatient admissions [1]. Many transferred patients are critically ill and high-risk who require facilities equipped to provide specialized services for their complex needs [2]. However, securing these services often requires travelling burdensome distances away from patients' homes and communities of support. Furthermore, these patients often arrive with poorly defined goals of care and experience unfavorable outcomes including higher rates of mortality [3–6]. The challenge is compounded by the fact that the transferred patients and families frequently do not understand the severity of illness, leading to unrealistic expectations and a potentially false sense of hope [7]. These circumstances impose significant burdens with unclear benefits, thus increasing the chance of rendering low value care [8].

Current practice leaves risk assessment to the ad-hoc judgment of bedside clinicians, suggesting an opportunity for better care coordination by more systematic procedures that should inform shared decisions about transferring critically ill patients or preparing the referral center to manage patients in line with patient preferences. This gap in coordination of patient care is largely due to the paucity of real-time tools to rapidly risk-stratify patients at the time of transfer. There are several mortality prediction tools used in ICU and other hospital settings. For example, the Acute Physiology, Age, Chronic Health Evaluation (APACHE) [9] and Sequential Organ Failure Assessment (SOFA) scores [10] are validated mortality predictors in the ICU, but because these tools are not feasibly implemented at the point of care, they have limited utility. Available admission mortality predictors have variable degrees of performance, and some are adapted for specific patient populations (*e.g.* surgical patients, geriatric trauma patients), limiting their utility for disease-agnostic acute settings [11–16]. Of the tools available, the 3-item quick SOFA (qSOFA) is the most feasible bedside mortality predictor with its rapid assessment in real-time. To date, it has been predominately tested in patients with infection and has shown modest discrimination and unclear predictive power [17]. Taken together, existing tools either fall short or are understudied in regard to reliable prediction of recovery from an acute episode in high-risk patients in real-time.

Based on the limitations of existing tools, the clinical leadership of a large, multi-hospital healthcare system defined a pressing need for a novel tool that could identify high-risk patents

at the time and point of transfer. The overarching goal of this quality improvement project was to provide needed information to front line physicians to inform shared decisions about the highest risk patients, directing additional resources to these patients to ensure that the plan of care was consistent with their values and goals. In this study, we describe how we use guided machine learning to develop and validate a tool called "SafeNET" (Safe Nonelective Emergent Transfers) that predicts expected mortality at the time of transfer based on variables available to bedside clinicians. We compared SafeNET's predictive capabilities with qSOFA, the only published tool to our knowledge suitable to ascertain mortality risk in this transfer patient population.

## Methods

### Setting

This study was conducted at the University of Pittsburgh Medical Center (UPMC), a large multiple-hospital integrated healthcare system. Work was approved by UPMC's Quality Review Committee as a quality improvement project (QRC #2040). Data are reported according to SQUIRE 2.0 standards for quality improvement reporting excellence [18].

### Population for model development

We developed the SafeNET mortality risk tool using retrospective data from patients aged 18 or older who were transferred to a UPMC hospital during a 12-month period. Transfer status was derived from the admission source on billing data indicating that the patient was transferred from a hospital, hospice, skilled nursing, or other health care facility with an admission priority of direct emergency admit. For patients with multiple inpatient stays during this time period, one record was randomly selected. Patients were excluded if their discharge disposition indicated they were discharged against medical advice, eloped, or had an unknown destination.

### Independent and dependent variables

Tool development began with a focused review of available literature on mortality risk assessment models currently used in ICU and admission settings. This literature review yielded 8 relevant articles describing 7 tools predicting mortality [10–17]. These included a tool developed for use in emergency department triage in Vietnam [11], the Early Warning Score [12,13], Simple Clinical Score [14], Rapid Emergency Medicine Score [16], Worthing Physiologic Score [15], SOFA score [10], and quick SOFA Score [17]. From these validated risk models, we constructed an extensive list of 70 independent variables used in one or more of these models including demographics, vital signs, lab tests, functional status, comorbidities, therapeutic maneuvers (e.g., respiratory support or blood product transfusion) and code status (S1 Table). We then queried UPMC billing data and the inpatient electronic health record (EHR) to determine if these variables were recorded by the receiving facility, focusing only on values recorded within 3 hours of transfer to most closely approximate the patient's condition at the time of transfer. After reviewing which variables were reliably available within this timeframe, we retained 54 of the 70 variables. If more than one value was recorded during this time, we selected the value closest to the time of admission. Functional status was assessed with the Boston University Activity Measure for Post-Acute Care (AM-PAC) Short Form "6-Clicks" [19]. We further linked each record to the patient's vital status and date of death, if deceased, using a proprietary file maintained by UPMC that combines the social security death index with other sources to render the best available record of vital status. We used the date of death to

calculate the dependent variable of In-hospital mortality, which was the primary outcome. We additionally examined secondary endpoints of mortality occurring 30 days and 90 days from the date of admission as well as a composite outcome for patients who survived, but who were transitioned to either hospice care or made "comfort measures only" during that admission.

## Guided machine learning modeling

We developed SafeNET using Gradient boosting because of its ability to automatically incorporate all variable interactions, account for any amount of missing information, and easily rank variables in terms of their predictive performance. We randomly divided the data into training (80%) and test (20%) sets to first develop the model and then test internal validity, respectively. Differences in training and test sets were examined with likelihood ratios, chi-squared and Student's t-tests for categorical and continuous variables, respectively. Three separate stochastic gradient boosting (XGBoost) algorithms with boosted decision trees and logarithmic loss function were employed on the training set to determine the influence and rank of variables for predicting each of the three mortality outcomes (in-hospital, 30-day and 90-day mortality) and CMO status. In order to focus on variables reliably available at the time of transfer, we modeled only those variables that were either (a) part of the Elixhauser Comorbidity Index [20] and thus reliably applied to all inpatient records or (b) for which at least 50% of our sample had valid values recorded within 3 hours of arrival. The model was then internally validated by using the same seed and running the algorithm a single time. Model discrimination was assessed with the c-statistic (e.g., area under the Receiver Operating Characteristic Curve); differences between training and test set c-statistics were assessed using the methods described by DeLong, et al. [21]. Model calibration was assessed with Spiegelhalter's z-test and by plotting observed and expected results across the range of risk. For each model, the variables were ranked according to importance and reviewed by a clinician with expertise in risk stratification (DEH) to select a limited set of variables for a parsimonious model that could feasibly be collected at the bedside yet capture most of the predictive power of the full model. The number of trees was set to 1,000 to tune model hyperparameters and subsequently monitor model performance.

In addition to calculating each patient's SafeNET score, we also calculated quick SOFA (qSOFA) scores on patients in the test set for whom all necessary information was available (respiratory rate, systolic blood pressure, Glasgow Coma Score). A Spearman correlation was used to assess the association between the two models. C-statistics, ROC curves, and calibration plots were obtained to compare the overall effectiveness of SafeNET and qSOFA scores to predict in-hospital mortality. All statistical analyses were conducted using R (Version3.4.1 "Single Candle"; R Core Team, 2017, Vienna, Austria [22]).

## Implementation and validation

After developing and internally validating SafeNET in the retrospective sample, we sought to determine SafeNET's validity and feasibility for prospective use in real time at the point of care. We built SafeNET as a web-based application that was easily accessible from any browser behind the UPMC firewall (**S1 Fig**). The application guided users to enter as many of the variables as were immediately available and then generated the predicted risk of each outcome by running a cloud-based instance of R pre-loaded with the gradient boosting algorithm. After securing support from the Vice President of Medical Affairs and the Chief Nursing Officer at each of three pilot hospitals within the UPMC system, a variety of training opportunities were made available to teach bedside nurses how to use the SafeNET tool. The UPMC Medcall team, which coordinates all transfers into and between UPMC facilities, was also enlisted.

After completing this academic detailing, we began piloting the prospective use of SafeNET with the following workflow: (1) When a bedside nurse from a piloting facility contacted Medcall to request a transfer to a higher level of care, the Medcall agent initiated the transfer process as usual, but asked the bedside nurse to complete the SafeNET score while they waited for bed assignment; (2) the nurse then accessed the web based tool, recorded patient identifiers and generated a SafeNET score which s/he (3) reported to the Medcall agent at the time that the agent connected the bedside nurses of the sending and receiving facilities for the typical sign-out discussion. The pilot was limited to patients older than 18 years transferred to a single, tertiary care hospital from select emergency department or intensive care units at one of the 3 UPMC referring hospitals. We excluded patients transferred for myocardial infarction (both those with and without ST-segment elevation) or maternal labor and delivery so as not to disturb existing transfer algorithms for these patient populations. Daily transfers were monitored by an implementation specialist (MKW) who made regular audit and feedback reports to participating facilities regarding their compliance with the SafeNET initiative as defined by the proportion of patients transferred for whom a SafeNET score was recorded. If compliance lagged, MKW offered additional academic detailing to lagging sites. We examined feasibility with compliance rates, data missingness, and the time to administer the tool. Prospective validation was assessed with c-statistics, calibration plots, Spiegelhalter's z, sensitivity, specificity, and both positive and negative predictive values for in-hospital, 30-day, and 90-day mortality.

## Results

### SafeNET development

A total of 20,985 patients were identified as transfer patients at a UPMC hospital between July 2017 and June 2018. Of these patients, 10,696 (51.0%) were directly admitted as inpatients whereas 10,289 (49.0%) were admitted as inpatients via the emergency department. Demographically, patients averaged 65 years of age, and the cohort was 52.0% male and 82.9% white race (**Table 1**). The patients had a mean Elixhauser Comorbidity Index of 4.2 conditions, and common Major Diagnostic Categories (MDC) of their inpatient stays were Diseases and Disorders of the Circulatory System or Nervous System (**Table 1**). There were 1,937 (9.2%) in-hospital mortalities, 2,884 (13.7%) 30-day mortalities, 3,899 (18.6%) 90-day mortalities, and 1,944 (9.3%) transitions to hospice or CMO among the transfer patients (**Table 1**). No differences were detected in patient demographics or characteristics between the training (N = 16,788) and test (N = 4,197) sets (**Table 1**).

After building models for each outcome using all 54 retained independent variables, the variables were ranked according to importance in predicting the outcome (**S2 Table**). Across the 4 outcomes, the top variables of importance included patient age, AM-PAC Activity score, blood urea nitrogen (BUN) levels, AM-PAC Mobility Score, fluid and electrolyte disorders, body temperature, mechanical ventilation, albumin, glucose, heart rate, systolic blood pressure, platelet count, and white blood cell count. There was a significant positive correlation between AM-PAC Activity Score and AM-PAC Mobility Score, so we retained the former to alleviate redundancy. Similarly, BUN and creatinine levels were highly correlated, so we included only BUN levels in the final model. Moreover, we chose to include a composite measure of cancer status (both metastatic cancer and solid tumor without metastases), both of which ranked highly in importance. These 14 variables were used to build the parsimonious model. **Table 2** describes the proportion of patients with data for each of these variables along with estimates of central tendency. With the exception of albumin levels, data were available for all variables in >90% of the cases. In the event that one or more variables were missing for any individual patient in the dataset, the SafeNET score was computed based only on the

**Table 1. Transfer patient characteristics, and mortality rates for the development phase of SafeNET.**

| Patient Demographics | Development | | | Prospective Validation |
| --- | --- | --- | --- | --- |
| | Overall | Training | Testing | Validation |
| | N = 20985 | N = 16788 | N = 4197 | N = 275 |
| Mean Age ± SD | 65.0 ± 18.1 | 65.0 ± 18.1 | 64.6 ± 18.1 | 59.1 ± 17.7 |
| Sex, vol (%) | | | | |
| Male | 10904 (52.0) | 8768 (52.2) | 2136 (50.9) | 141 (51.3) |
| Female | 10081 (48.0) | 8020 (47.8) | 2061 (49.1) | 134 (48.7) |
| Race, vol (%) | | | | |
| White | 17399 (94.3) | 13873 (94.2) | 3526 (94.8) | 239 (86.9) |
| Black | 976 (5.3) | 798 (5.4) | 178 (4.8) | 33 (12.0) |
| Other | 78 (0.4) | 64 (0.4) | 14 (0.4) | 3 (1.1) |
| **Patient Characteristics** | | | | |
| Elixhauser Comorbidity, mean ± SD | 4.2 ± 2.6 | 4.2 ± 2.6 | 4.3 ± 2.6 | 4.3 ± 2.7 |
| Major Diagnostic Category, vol (%) | | | | |
| Circulatory System | 4084 (19.5) | 3245 (19.3) | 839 (20.0) | 33 (12.0) |
| Nervous System | 3561 (17.0) | 2854 (17.0) | 707 (16.8) | 75 (27.3) |
| Orthopedic System | 2054 (9.8) | 1625 (9.7) | 429 (10.2) | 27 (9.8) |
| Respiratory System | 2040 (9.7) | 1654 (9.9) | 386 (9.2) | 21 (7.6) |
| Digestive System | 1915 (9.1) | 1527 (9.1) | 388 (9.2) | 24 (8.7) |
| Infectious & Parasitic Diseases | 1794 (8.5) | 1456 (8.7) | 338 (8.1) | 19 (6.9) |
| Hepatobiliary System, Pancreas | 975 (4.6) | 757 (4.5) | 218 (5.2) | 17 (6.2) |
| Kidney & Urinary Tract | 854 (4.1) | 673 (4.0) | 181 (4.3) | 0 (0) |
| Ear, Nose, Mouth, and Throat | 279 (1.3) | 231 (1.4) | 48 (1.1) | 10 (3.6) |
| Other | 3429 (16.3) | 2766 (16.5) | 663 (15.8) | 49 (17.8) |
| **Mortality** | | | | |
| In-Hospital | 1937 (9.2) | 1529 (9.1) | 408 (9.7) | 25 (9.1) |
| 30-Day | 2884 (13.7) | 2289 (13.6) | 595 (14.2) | 27 (13.2) |
| 90-Day | 3899 (18.6) | 3083 (18.4) | 816 (19.4) | 43 (15.6) |
| Transition to hospice/CMO | 1,944 (9.3%) | 1,527 (9.1%) | 417 (9.9%) | ———————————— |

All p-values comparing differences between the training and test sets were >0.1.

available data (e.g., no imputation of missing values). This approach mimics real-world conditions and leverages gradient boosting's ability to render the best possible prediction given the available data with any amount of missing data.

As expected, the full 54-variable model had the best discrimination across all mortality outcomes with c-statistics of 0.903 (95% CI: 0.891–0.916), 0.877 (95%CI: 0.864–0.890), and 0.869 (95%CI: 0.857–0.881) for in-hospital, 30-day and 90-day mortality, respectively. As indicated by a Spiegelhalter's Z-test p-value $\geq$ 0.05 consistent with no significant difference between observed and predicted values across the range of risk, calibration of the 54-variable model was good for predicting in-hospital and 30-day mortality (**Fig 1A & 1B**) but was less accurate for 90-day mortality (Spiegelhalter's Z-test p-value<0.05; **Fig 1C**). The parsimonious 14-variable model demonstrated better calibration for predictions of in-hospital and 30-day mortality because, unlike the 54-variable model, it was tuned to adjust for hyperparameters (Spiegelhalter's Z-test P-value$\geq$0.05; **Fig 1E & 1F**). However, calibration of the 14-variable model was less accurate for predicting 90-day mortality (Spiegelhalter's Z-test p-value<0.05; **Fig 1G**). After tuning, the parsimonious models achieved most of the full model's discrimination with c-

**Table 2. Completeness of data and respective values for patient variables that have the most important variables predictive of mortality by mortality status.**

| Variable | Non-Mortalities Total Vol = 16063 | | In-Hospital Mortalities Total Vol = 1937 | | 30-Day Mortalities Total Vol = 2884 | | 90-Day Mortalities Total Vol = 3899 | |
|---|---|---|---|---|---|---|---|---|
| | N (%) | Value | N (%) | Value | N (%) | Value | N (%) | Value |
| Patient Age ± SD | 16063 (100) | 62.4±18.2 | 1937 (100) | 71.7±15.7 | 2884 (100) | 73.4±15.2 | 3899 (100) | 73.3±14.9 |
| AM-PAC Activity Score*± SD | 15441 (96.1) | 19.1±5.8 | 1795 (92.7) | 11.6±6.3 | 2711 (94.0) | 12.5±6.3 | 3689 (94.6) | 13.4±6.4 |
| BUN (mg/dL) ± SD | 15649 (97.4) | 21.0±16.5 | 1859 (96.0) | 37.2±25.4 | 2784 (96.5) | 35.5±24.7 | 3782 (97.0) | 34.4±24.3 |
| Albumin (g/dL) ± SD | 10466 (65.2) | 3.2±0.6 | 1655 (85.4) | 2.7±0.7 | 2403 (83.3) | 2.8±0.7 | 3204 (82.2) | 2.8±0.7 |
| Blood Glucose (mg/dL) ± SD | 15137 (94.2) | 133.8±66.8 | 1848 (95.4) | 163.7±94.0 | 2750 (95.3) | 155.2±85.5 | 3726 (95.6) | 150.6±82.2 |
| Body Temperature (˚C) ± SD | 16019 (99.7) | 36.7±1.3 | 1890 (97.6) | 36.3±2.0 | 2835 (98.3) | 36.5±2.1 | 3849 (98.7) | 36.5±1.8 |
| Cancer with Metastases, vol (%) | 16063 (100) | 378 (2.4) | 1937 (100) | 141 (7.3) | 2884 (100) | 345 (12.0) | 3899 (100) | 536 (13.8) |
| Solid Tumor w/o Metastasis, vol (%) | 16063 (100) | 834 (5.2) | 1937 (100) | 216 (11.2) | 2884 (100) | 460 (16.0) | 3899 (100) | 728 (18.7) |
| Mechanical Ventilation, vol (%) | 16000 (99.6) | 1065 (6.7) | 1903 (98.2) | 651 (34.2) | 2848 (98.8) | 701 (24.6) | 3861 (99.0) | 767 (19.9) |
| Platelets (x1,000/μL) ± SD | 15640 (97.4) | 223.7±96.0 | 1863 (96.2) | 204.2±109.9 | 2790 (96.7) | 207.8±109.4 | 3793 (97.3) | 212.9±114.1 |
| White Blood Cells (x1,000/μL) ± SD | 15656 (97.5) | 11.1±8.9 | 1866 (96.3) | 15.6±14.6 | 2795 (96.9) | 14.9±18.1 | 3800 (97.5) | 14.0±16.1 |
| Fluid/Electrolyte Disorders, vol (%) | 16063 (100) | 4393 (27.4) | 1937 (100) | 1214 (62.7) | 2884 (100) | 1657 (57.5) | 3899 (100) | 2144 (55.0) |
| Systolic Blood Pressure (mm Hg) ± SD | 16023 (99.8) | 137.1±26.3 | 1901 (98.1) | 125.2±31.2 | 2847 (98.7) | 127.6±30.3 | 3861 (99.0) | 128.5±29.4 |
| Heart Rate (beats/min) ± SD | 16024 (99.8) | 84.8±22.4 | 1924 (99.3) | 93.6±24.9 | 2870 (99.5) | 92.5±23.4 | 3884 (99.6) | 91.4±22.7 |

*Lower scores equate to lower levels of function.

statistics of 0.876 (95% CI: 0.860–0.891, 0.855 (95% CI: 0.840–0.870), and 0.848 (95%CI: 0.834–0.861) for in-hospital, 30-day and 90-day mortality, respectively.

We also investigated the ability of SafeNET to predict patients whose care would transition to comfort measures only or hospice during the hospital stay. Both the full 54-variable model and 14-variable model showed good discrimination with c-statistics of 0.867 (95% CI: 0.852–0.881) and 0.840 (95% CI: 0.822–0.857), respectively. However, calibration plots indicated that both models were less accurate at predicting CMO/hospice care transition (Spiegelhalter's Z-test p-value<0.05 for both models) (**Fig 1D and 1H**) with the model overpredicting the risk of the composite CMO/Hospice transition. However, these models were not tuned for hyperparameters.

We next compared the capabilities of SafeNET versus qSOFA at predicting in-hospital mortality. Among the 4,197 patients randomly selected to be a part of the test cohort (**Table 1**), 2,260 of these patients had sufficient clinical information to calculate a qSOFA score. We found a significant positive association between SafeNET and qSOFA scores (Spearman Correlation Coefficient = 0.536; p<0.001). Evaluation of ROC curves indicated that SafeNET had superior discrimination to qSOFA (c = 0.836, 95% CI: 0.814–0.857 vs. 0.713, 95% CI: 0.686–0.739, p<0.001 **Fig 2**).

## SafeNET validation

The parsimonious 14-variable model was chosen to build a web-based SafeNET tool (**S1 Fig**). This tool was successfully implemented in December 2018 with the goal of recording a Safe-NET score for 80% of transferred patients (e.g., 80% compliance). From December 2018 to June 2019, 679 patients were transferred from one of 3 referring UPMC hospitals to a single, tertiary receiving hospital. Overall, SafeNET scores were obtained for 429 patients (e.g., 63.2% compliance), but compliance improved month over month from 41.3% in December to 77.5% in June with the highest performing hospital achieving compliance of 83.3% (**Figs 3A and 4B**). The median time to complete the survey was 4.5 min (IQR = 3.2 min), and fell over time,

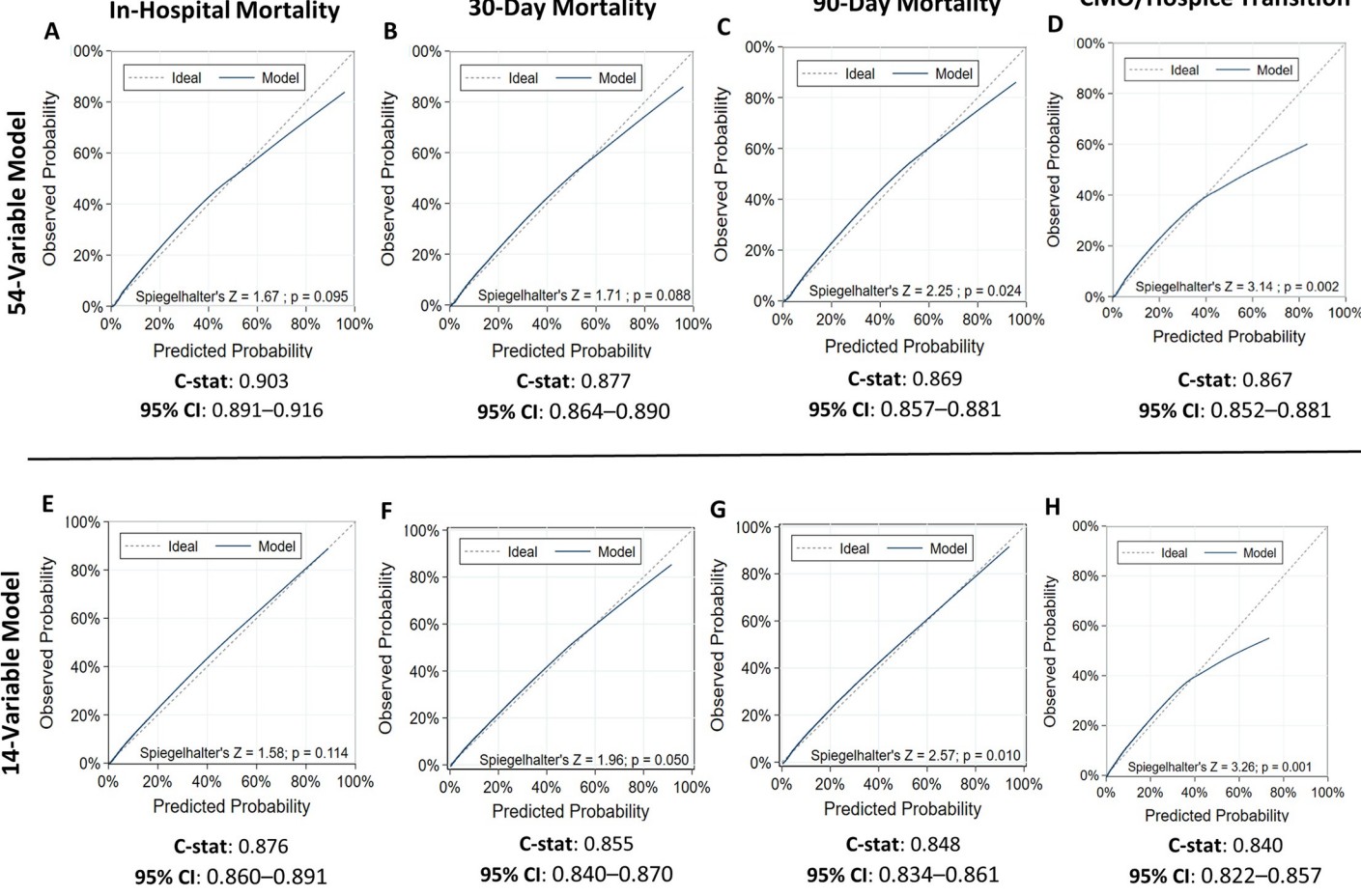

**Fig 1. Comparison of discrimination and calibration during model development.** Calibration plots and discrimination outcomes of guided machine learning are displayed for models using all 54 patient variables (A-D) vs. the top 14 most important variables (E-H) to predict in-hospital, 30-day, and 90-day mortality and CMO/hospice transition among hospital transfer patients.

beginning at 5.1 min (IQR = 4.9) in December (or December/January) to 4.4 min (IQR = 3.1min) in June.

Of the 429 recorded SafeNET scores, 275 (64.1%) were successfully linked back to the EHR and vital status. Data linkage was limited by technical constraints due to HIPAA compliance that required collecting SafeNET data without patient identifiers, thereby necessitating a cross-walk table that relied on manual data entry from the bedside nurse. Demographically, the validation cohort was similar to those in the development cohort with a mean age of 59.1, 51.3% male, and 86.9% white race (**Table 1**). Moreover, most patients fell under MDCs of Disorders of the Nervous system (28.0%), Circulatory System (12.3%), or Musculoskeletal System (10.1%). Among these patients, in-hospital, 30-day, and 90-day mortality occurred in 25 (9.1%), 27 (13.2%), and 43 (15.6%) patients, respectively (Table 1). Data entry was complete for all 14 variables on the SafeNET tool for 36% of SafeNET scores. Variables with the lowest completion rates included albumin levels (61.1%), cancer history (74.5%), AM-PAC Activity Score (82.6%), and Fluid and Electrolyte Disorders (83.6%). The prospectively calculated Safe-NET score predicted each outcome as expected with c-statistics only slightly lower those observed in the development sample (c = 0.836, 95%CI: 0.751–0.921; c = 0.827, 95% CI: 0.743–

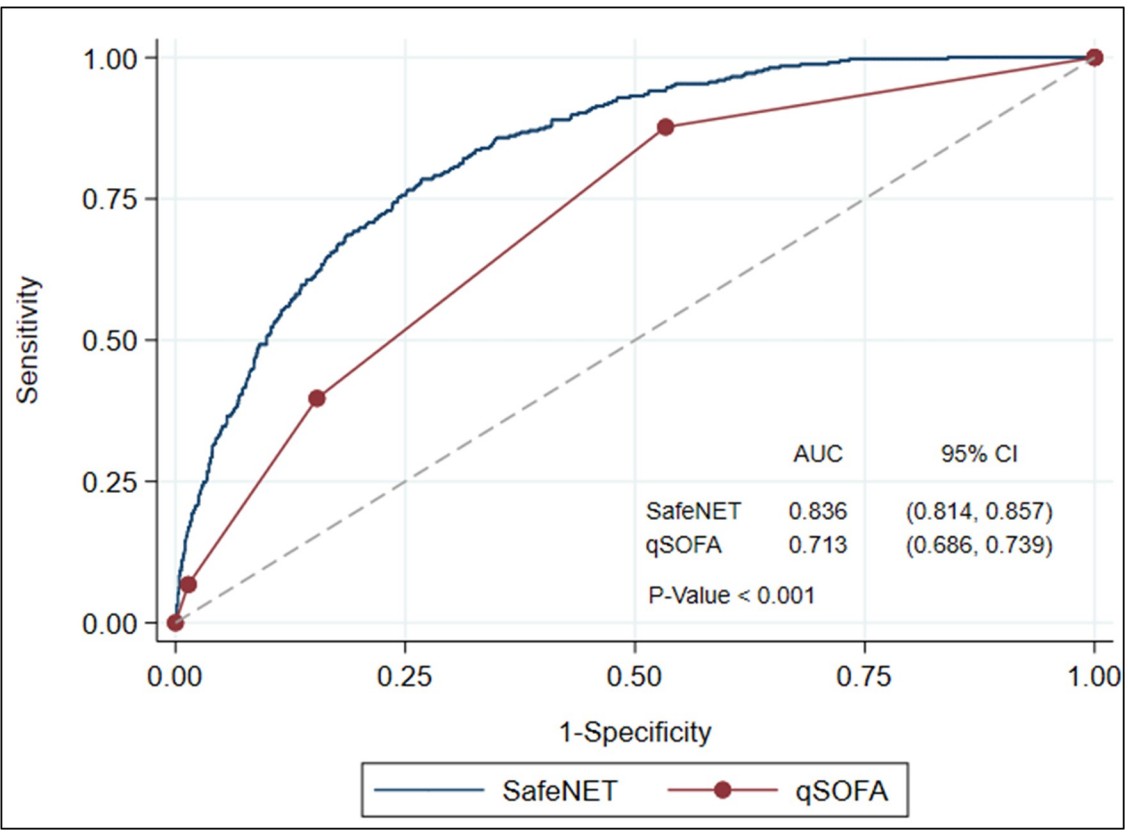

**Fig 2. Comparison of receiver operating characteristic (ROC) curves for SafeNET and qSOFA.** The ROC curves and discrimination of the SafeNET (Safe Nonelective Emergent Transfers) and qSOFA (quick Sequential Organ Failure Assessment) models are displayed.

0.911; c = 0.814, 95% CI: 0.744–0.884 for in-hospital, 30-day, and 90-day mortality, respectively) (**Fig 4A–4C**). Calibration was accurate for in-hospital, 30-day, and 90-day mortality outcomes (Spiegelhalter's Z-test P-value>0.05; **Fig 4D–4F**) and generally under-predicted observed in-hospital mortality for event rates greater than 40%. At a threshold of at least 20% predicted in-hospital mortality, sensitivity was 56% (14/25), specificity was 93% (233/250), and the positive predictive value (PPV) was 45% (14/31) (**Table 3**). Test statistics were similar for 30-day (sensitivity = 64.3%, specificity = 87%, PPV = 36%) and 90-day mortality (sensitivity = 67%, specificity = 81%, PPV = 39%) (**Table 3**).

## Discussion

Although interhospital transfer of critically ill patients is a common occurrence, there is no standardized process to assess these patients' risk for poor outcomes and communicate that crucial information at the initial point-of-care. This often leads to care plans misaligned with patient values and poor outcomes. As part of a quality improvement initiative, our institution developed SafeNET, a robust mortality prediction tool with the sensitivity to accurately risk-stratify critically ill patients at the time of hospital transfer. Feasibility of SafeNET was demonstrated by increasing compliance rates and ability to complete and automatically generate predicted mortality within minutes at the point of care. Moreover, performance of SafeNET was validated in a prospective cohort which demonstrated discrimination and calibration on par

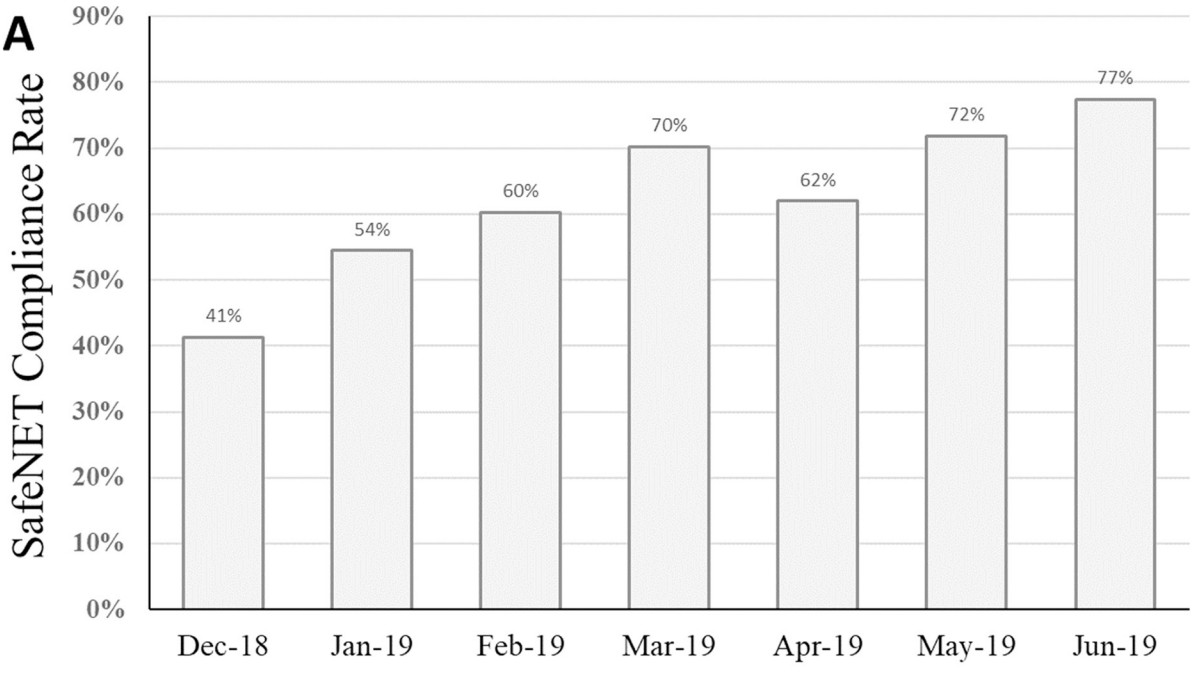

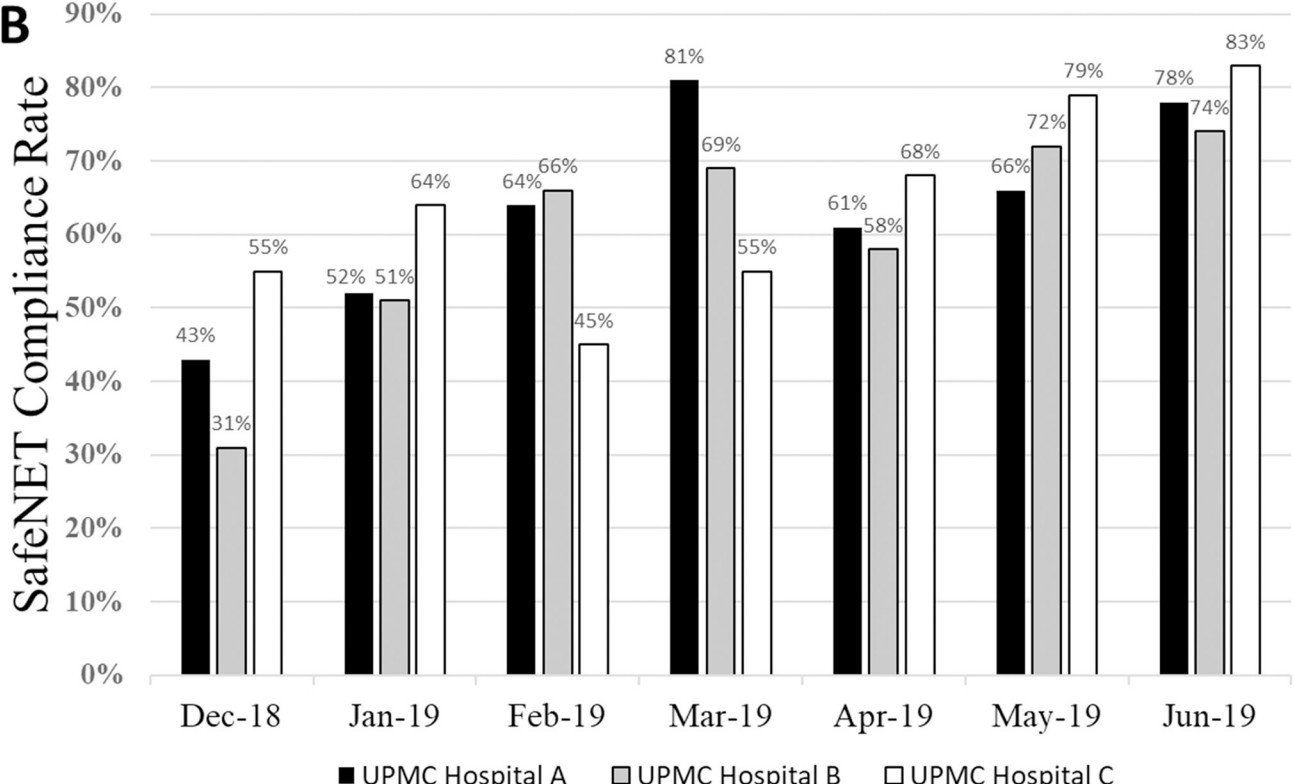

**Fig 3. SafeNET compliance over time by hospital facility.** Compliance rates are displayed during a 7-month pilot period (A) and broken down by participating facility (B).

# SafeNET Validation

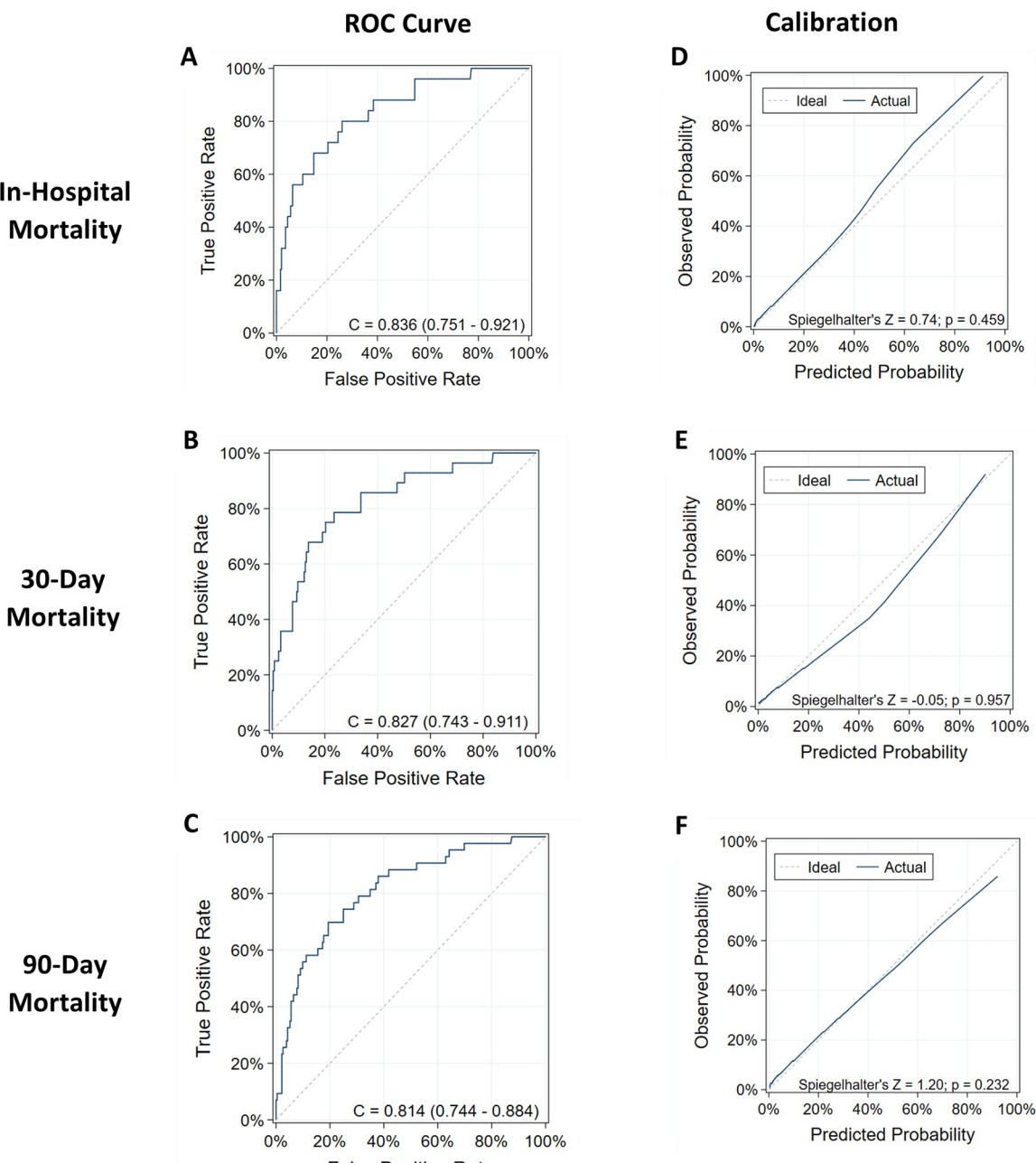

**Fig 4. Discrimination and calibration of SafeNET models for predicting mortality.** Validation of SafeNET for predicting in-hospital (A,D), 30-day (B,E), and 90-day (C,F) mortality among transfer patients is shown.

with that produced in the development phase, particularly for predictions of in-hospital and 30-day mortality.

The SafeNET tool has several advantages over current mortality prediction tools developed for admission and ICU settings. First, SafeNET uses only 14-variables that are readily available at the time of presentation, taking only 4 minutes to complete at the bedside. By contrast,

**Table 3. Sensitivity, specificity, and predictive values of prospectively measured SafeNET score at a threshold of 20% mortality.**

| Outcome | Prevalence | Sensitivity | Specificity | PPV | NPV | Accuracy |
|---|---|---|---|---|---|---|
| In-Hospital Mortality | 25/275 (9.1%) | 14/25 (56.0%) | 233/250 (93.2%) | 14/31 (45.2%) | 233/244 (95.5%) | 247/275 (89.8%) |
| 30-Day Mortality | 28/275 (10.2%) | 18/28 (64.3%) | 215/247 (87.0%) | 18/50 (36.0%) | 215/225 (95.6%) | 233/275 (84.7%) |
| 90-Day Mortality | 43/275 (15.6%) | 29/43 (67.4%) | 187/232 (80.6%) | 29/74 (39.2%) | 187/201 (93.0%) | 216/275 (78.5%) |

existing tools rely on clinical data collected over the course of 24 hours, which cannot accommodate the clinically acute need to identify high-risk patients during the critical timeframe interhospital transfer [23–27]. Moreover, some of the existing tools require input of a large number of variables, which may be difficult to obtain and time consuming to complete [23,28].

Second, although rapid, brief and thus feasible for bedside assessment, SafeNE's predictive ability outperforms qSOFA, the only other tool to our knowledge proven feasible for bedside assessment, and it does so in a diverse population agnostic of disease process. The qSFOA was designed as an abbreviated version of the SOFA score and was shown to outperform SOFA in non-ICU settings at predicting in-hospital mortality in patients with infection and suspected sepsis [17]. It can be feasibly implemented at the bedside and requires only three clinical criteria including respiration rate, systolic blood pressure, and the Glasgow Coma Score [17]. However, the qSOFA model was built as a short-term mortality prediction tool using patients with infection, and reports on predictive capabilities in more generalized non-infected patient populations have varied findings [29–31]. In the current study, we found that SafeNET is superior to qSOFA at prediction of in-hospital mortality (c = 0.836 vs c = 0.713, p < .001), and that it does so agnostic of disease process in a diverse sample. This is likely due to the more complex nature of our 14-variable model, but we have shown that these variables are readily obtainable at the bedside with minimal disruption to clinical workflow.

Third, SafeNET is robust to missing data, effectively making risk predictions with whatever data is readily at hand. Previous mortality prediction models had to exclude records for missing data [16]. However, the gradient boosting algorithm used to develop SafeNET is able to account for missingness so that not all variables are needed to make a prediction. In the bedside setting, a degree of missingness of clinical variables is expected due to timing issues and difficulty obtaining information, but the SafeNET tool was able to provide strong discrimination and calibration even in a setting where only 54% of predictions were made with at least one missing variable.

Fourth, the manual data entry format of the online SafeNET tool is implementable at any bedside with ready access to an internet connection. This is critically important considering that at our center, approximately 75% of patients transferred into our hospitals come from outside institutions with isolated electronic records. As such, even if SafeNET were automated for within-system transfers based on data extant in the electronic record, a manual alternative would be required for the majority of patients arriving from outside institutions where a link to the SafeNET tool could be easily delivered via a variety of electronic media.

Finally, our data demonstrate that SafeNET has good discrimination for predicting transition to comfort measures only or hospice care, although this prediction was not as well calibrated as the model for predicting mortality as we did not tune this for hyperparameters. There are limited resources for predicting patients whose care will be transferred to hospice or comfort measures only. The Hospital End-of-Life Prognostic Score (HELPS) predicts the aggregate outcome of in-hospital mortality and discharge to hospice using variables such as patient demographics, resuscitation status, nutrition status, and comorbidities, and its retrospective development demonstrated discrimination (C = 0.866) comparable to SafeNET [32].

However, it has not been prospectively validated at the bedside as done here, and it relies on the calculation of potentially cumbersome and time consuming subscores such as the Inpatient Physiologic Failure Score that itself requires an assessment of 12 variables including vital signs, blood chemistry and consciousness. Other tools have been developed with the specific goal of identifying patients who would benefit from palliative care, but these are tailored for more specific populations such as cancer patients [33]. Predicting these transitions to end-of-life care are especially important, perhaps even before transferring to a higher level of care, because they can trigger conversations clarifying patient goals and preferences, but future work is necessary to determine if and how these end-of-life transition predictions translate to more goal-concordant care plans.

There are several limitations to employing SafeNET as a point-of-care mortality risk model. First, data is restricted to a single, multi-hospital healthcare system and findings may not generalize to other settings. Second, some of the information that feeds SafeNET's algorithms may be subject to bias (*e.g.* rater bias with completing the AM-PAC Activity score) and bedside assessment of comorbidities is likely different that the post-facto administrative ICD-10 coding on which the models were developed. However, model discrimination and calibration performed as expected, and the real-time calculation proved feasible with modest effort of bedside clinicians. Third, manual data entry is required. Future automation may further expedite implementation for patients already within a hospital system's data infrastructure, but manual data entry is feasible for immediate implementation and may actually facilitate calculation when patients are transferred from an outside hospital that does not share data infrastructure.

In conclusion, the development of SafeNET provides an objective and systematic way to risk stratify patients at the time of hospital transfer; its use and generalizability across other health systems and settings remains to be determined and is a focus of future work. SafeNET is not meant to supersede clinical judgement, but rather it is intended as a means to trigger a pause so that clinicians are better prepared to inform high-risk patients (or their surrogates) of the severity of their illness and address goals of care when they arrive at the receiving facility, or in some cases, before transferring patients outside their communities of support. Patients who are aware of their condition and participate in conversations with physicians about their values tend to receive care that is consistent with these care preferences [34]. Moreover, both patient and family satisfaction are significantly improved among patients whose care included physician-directed advance care planning [35]. Ongoing work detailing the implementation process to mainstream SafeNET are necessary next steps to facilitate these efforts.

## Supporting information

**S1 Fig. Web-based SafeNET survey completed at patients' bedsides.**
(DOCX)

**S1 Table. List of 70 patient variables considered during the development of SafeNET.**
(DOCX)

**S2 Table. Rank of variable importance of the 54 variables.**
(DOCX)

## Acknowledgments

The authors would like to thank Dr. Jonas T. Johnson and Dr. Steven D. Shapiro for their insights in generating the ideas behind the establishment of the SafeNET tool. The authors would also like to thank Dr. David L. Burwell, Dr. Vincent J. Silvaggio, Mary Barkhymer, Dawndra Jones, Laura Gailey Moul, Stacey-Ann Okoth, and Colleen Sullivan for their efforts

to help coordinate implementation of SafeNET across pilot sites within the UPMC system and Adam Yee and Jeremy Wells for their work in building the SafeNET tool as a web-based application.

## Author Contributions

**Conceptualization:** Johanna E. Bellon, Jeffrey D. Borrebach, Tamra E. Minnier, Joel B. Nelson, Daniel E. Hall.

**Data curation:** Johanna E. Bellon, Andrew Bilderback, Jeffrey D. Borrebach, Jacob C. Hodges, Daniel E. Hall.

**Formal analysis:** Stefanie C. Altieri Dunn, Johanna E. Bellon, Andrew Bilderback, Jeffrey D. Borrebach, Jacob C. Hodges.

**Investigation:** Mary Kay Wisniewski, Daniel E. Hall.

**Methodology:** Andrew Bilderback, Mary Kay Wisniewski, Daniel E. Hall.

**Project administration:** Mary Kay Wisniewski.

**Resources:** Matthew E. Harinstein, Tamra E. Minnier, Joel B. Nelson.

**Supervision:** Johanna E. Bellon, Tamra E. Minnier, Joel B. Nelson, Daniel E. Hall.

**Writing – original draft:** Stefanie C. Altieri Dunn, Mary Kay Wisniewski, Daniel E. Hall.

**Writing – review & editing:** Stefanie C. Altieri Dunn, Johanna E. Bellon, Andrew Bilderback, Jeffrey D. Borrebach, Jacob C. Hodges, Mary Kay Wisniewski, Matthew E. Harinstein, Tamra E. Minnier, Joel B. Nelson, Daniel E. Hall.

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
