## [Decision Letter · Decision Letter 0]

2 Nov 2020

PONE-D-20-27880

SafeNET: Initial development and validation of a real-time tool for predicting mortality risk at the time of hospital transfer to a higher level of care

PLOS ONE

Dear Dr. Hall,

Thank you for submitting your manuscript to PLOS ONE. After careful consideration, we feel that it has merit but does not fully meet PLOS ONE’s publication criteria as it currently stands. Therefore, we invite you to submit a revised version of the manuscript that addresses the points raised during the review process.

We look forward to receiving your revised manuscript.

Kind regards,

Ramesh Kumar, PhD

Academic Editor

PLOS ONE

Journal Requirements:

Reviewers' comments:

Reviewer's Responses to Questions

**Comments to the Author**

1. Is the manuscript technically sound, and do the data support the conclusions?

Reviewer #1: Yes

Reviewer #2: Yes

2. Has the statistical analysis been performed appropriately and rigorously? 

Reviewer #1: Yes

Reviewer #2: Yes

3. Have the authors made all data underlying the findings in their manuscript fully available?

Reviewer #1: Yes

Reviewer #2: Yes

4. Is the manuscript presented in an intelligible fashion and written in standard English?

Reviewer #1: Yes

Reviewer #2: Yes

5. Review Comments to the Author

Reviewer #1: The manuscript is well written with all the components addressed satisfactorily. The data has been analysed and interpreted appropriately. English language used in the manuscript is good for grammar. The SafeNET algorithm used for coming up with a real-time tool to improve the good patient outcomes is an innovative idea.

Reviewer #2: The manuscript titled "Development and validation of a real-time tool for predicting mortality risk at the time of hospital transfer", is a tremendous contribution of the authors towards an empirically deficient area of clinical practice. The contributions are understood to facilitate practitioners and clinicians in ascertaining the risk of mortality across specified time frames during hospital transfer of critically ill patients.

It is suggested that the authors may slightly modify their background section to align more with the qSOFA and SafeNet tools, as currently it narrates about the existing tools but lesser mention of one of the prime tools that have been assessed in the study.

In methods section, the authors have narrated the adopted methodologies and approaches at length, and no major modifications are suggested. However, it would useful to have some more description about the process of independent variables selection from the existing tools and prioritization for clinical findings/observations.

In the result section, it would be useful to relate the process of testing the tool with the clinical outcomes in the narrative as well. The tables and illustrations detail the statistical outputs well, however the narrative relates less with the 30 days and 90 days mortality aspects as observed in the study.

There is reasonable attribution of study findings with other referenced materials in the discussion session, however, the conclusions mention about the deployment steps, rather than testing out the application of tools in other settings as well.

6. PLOS authors have the option to publish the peer review history of their article (what does this mean?). If published, this will include your full peer review and any attached files.

Reviewer #1: **Yes: **Dr. Gul Muhammad Baloch

Reviewer #2: No

---

## [Author Response · Author response to Decision Letter 0]

18 Dec 2020

Reviewer #1

The manuscript is well written with all the components addressed satisfactorily. The data has been analyzed and interpreted appropriately. English language used in the manuscript is good for grammar. The SafeNET algorithm used for coming up with a real-time tool to improve the good patient outcomes is an innovative idea. 

We are humbled by the positive assessment of Reviewer #1 and are thankful for their time and thoughtful comments regarding the SafeNET tool. 

Reviewer #2 

The manuscript titled "Development and validation of a real-time tool for predicting mortality risk at the time of hospital transfer", is a tremendous contribution of the authors towards an empirically deficient area of clinical practice. The contributions are understood to facilitate practitioners and clinicians in ascertaining the risk of mortality across specified time frames during hospital transfer of critically ill patients. 

We thank Reviewer #2 for their generous evaluation and for recognizing the clinical importance of the need for such a tool as SafeNET.

It is suggested that the authors may slightly modify their background section to align more with the qSOFA and SafeNet tools, as currently it narrates about the existing tools but lesser mention of one of the prime tools that have been assessed in the study.

We modified the arrangement and content of the second paragraph of the introduction to provide more clarity and detail about qSOFA and its current limitations. We also added a sentence to the last paragraph of the introduction to explicitly mention that we directly compared SafeNET with qSOFA as this is the most logical tool available for comparison.

In methods section, the authors have narrated the adopted methodologies and approaches at length, and no major modifications are suggested. However, it would useful to have some more description about the process of independent variables selection from the existing tools and prioritization for clinical findings/observations.

A thorough search of the literature identified 8 articles that described 7 applicable mortality risk tools. We reviewed these in depth to extract a comprehensive list of 70 independent variables that had been identified in previous research as promising predictors for this purpose. We then used information available to us through billing and electronic health records to identify which of these variables were readily accessible within a clinically relevant timeframe. This brought us down to 54 independent variables that were tested in our gradient boosting models. We added these additional details to the Independent and Dependent Variables portion of the Methods section. 

In the result section, it would be useful to relate the process of testing the tool with the clinical outcomes in the narrative as well. The tables and illustrations detail the statistical outputs well, however the narrative relates less with the 30 days and 90 days mortality aspects as observed in the study.

We added in detailed results for the sensitivity, specificity, and positive predictive abilities for 30-day and 90-day outcomes. This information is in the last paragraph of the results section.

There is reasonable attribution of study findings with other referenced materials in the discussion session, however, the conclusions mention about the deployment steps, rather than testing out the application of tools in other settings as well.

We agree that the generalizability of SafeNET across different settings is unknown. We added a sentence to the last paragraph of the conclusion section to mention this limitation.

---

## [Decision Letter · Decision Letter 1]

25 Jan 2021

SafeNET: Initial development and validation of a real-time tool for predicting mortality risk at the time of hospital transfer to a higher level of care

PONE-D-20-27880R1

Dear Author,

We’re pleased to inform you that your manuscript has been judged scientifically suitable for publication and will be formally accepted for publication once it meets all outstanding technical requirements.

Kind regards,

Ramesh Kumar, PhD

Academic Editor

PLOS ONE

Additional Editor Comments (optional):

Reviewers' comments:

Reviewer's Responses to Questions

**Comments to the Author**

1. If the authors have adequately addressed your comments raised in a previous round of review and you feel that this manuscript is now acceptable for publication, you may indicate that here to bypass the “Comments to the Author” section, enter your conflict of interest statement in the “Confidential to Editor” section, and submit your "Accept" recommendation.

Reviewer #1: All comments have been addressed

Reviewer #2: All comments have been addressed

2. Is the manuscript technically sound, and do the data support the conclusions?

Reviewer #1: Yes

Reviewer #2: Yes

3. Has the statistical analysis been performed appropriately and rigorously? 

Reviewer #1: Yes

Reviewer #2: Yes

4. Have the authors made all data underlying the findings in their manuscript fully available?

Reviewer #1: Yes

Reviewer #2: Yes

5. Is the manuscript presented in an intelligible fashion and written in standard English?

Reviewer #1: Yes

Reviewer #2: Yes

6. Review Comments to the Author

Reviewer #1: The authors have responded positively and have made the amendments accordingly.

The paper may be accepted now for publication.

Reviewer #2: The authors have adequately addressed all comments in the revised version of the manuscript. All required questions have been answered and that all responses meet formatting specifications.

7. PLOS authors have the option to publish the peer review history of their article (what does this mean?). If published, this will include your full peer review and any attached files.

Reviewer #1: **Yes: **Dr Gul Muhammad Baloch

Reviewer #2: **Yes: **Ahsan Maqbool Ahmad

---

## [Editor Report · Acceptance letter]

29 Jan 2021

PONE-D-20-27880R1 

 SafeNET: Initial development and validation of a real-time tool for predicting mortality risk at the time of hospital transfer to a higher level of care 

Dear Dr. Hall:

I'm pleased to inform you that your manuscript has been deemed suitable for publication in PLOS ONE. Congratulations! Your manuscript is now with our production department. 

Kind regards, 

on behalf of

Dr. Ramesh Kumar 

Academic Editor

PLOS ONE